# Analysis of (p)ppGpp metabolism and signaling using a dynamic luminescent reporter

**Molly Hydorn**[1], **Sathya Narayanan Nagarajan**[1], **Elizabeth Fones**[2],
**Caroline S. Harwood**[2], **Jonathan Dworkin**[1]*

1 Department of Microbiology and Immunology, Vagelos College of Physicians and Surgeons, Columbia University, New York, New York, United States of America, 2 Department of Microbiology, University of Washington, Seattle, Washington, United States of America

* jonathan.dworkin@columbia.edu

**Data availability statement:** All Data are in the manuscript and/or Supporting information files

**Funding:** This work was supported by National Institutes of Health (https://www.nih.gov/) grant R35GM141953 to JD and United States Army Research Office (https://arl.devcom.army.mil/who-we-are/aro/) contract

## Abstract

As rapidly growing bacteria begin to exhaust nutrients, their growth rate slows, ultimately leading to the non-replicative state of quiescence. Adaptation to nutrient limitation requires widespread metabolic remodeling that is in part mediated by the phosphorylated nucleotides guanosine tetra- and penta-phosphate, collectively (p)ppGpp. We have developed a novel reporter of (p)ppGpp abundance in the Gram-positive bacterium *Bacillus subtilis* based on the recent identification of a riboswitch that binds (p)ppGpp and modulates transcription via regulation of a transcriptional terminator. Placement of an unstable reporter, firefly luciferase, downstream of the riboswitch allows for sensitive and dynamic assessment of (p)ppGpp. We first confirm that the reporter accurately reflects (p)ppGpp abundance in a variety of well-established conditions. We then proceed to use it to demonstrate the physiological importance of several mechanisms of regulation of (p)ppGpp metabolism previously observed only *in vitro* including allosteric interactions between (p)ppGpp synthesis enzymes and the hydrolytic activity of a (p)ppGpp synthetase. (p)ppGpp signaling has been implicated in the regulation of gene expression, and we demonstrate a close temporal association between gene expression and (p)ppGpp abundance, indicating a rapid, and therefore likely direct mechanism of (p)ppGpp dependent gene activation. Thus, this reporter provides a new, comprehensive analysis of (p)ppGpp signaling *in vivo* and offers the potential ability to sensitively monitor the temporal dynamics of (p)ppGpp abundance under diverse environmental conditions.

## Author summary

Most bacteria adapt to stressful conditions such as nutrient limitation by synthesizing a signaling molecule, known as ppGpp, that consists of a hyper-phosphorylated GTP. Synthesis of ppGpp affects most aspects of cellular

W911NF2110015 to CSH. MH acknowledg-
es support from the Columbia University
Graduate Training Program in Microbiology
and Immunology (National Institutes of
Health Allergy And Infectious Disease
grant T32AI106711) and the Training in
Cardiovascular Translational Research Training
Grant (National Institutes of Health, Heart Lung
and Blood Institute grant 5T32HL120826).
The funders had no role in study design, data
collection and analysis, decision to publish, or
preparation of the manuscript.

**Competing interests:** The authors have
declared that no competing interests exist.

physiology including replication, transcription and translation. We present here
a method that allows measurement of ppGpp abundance in living cells, greatly
facilitating investigation into ppGpp metabolism.

## Introduction

The adaptive response of bacteria to environmental changes can be relatively spe-
cific, as in the case of two-component signaling that involves a single transcription
factor regulating a limited number of genes. Alternatively, an adaptive response can
be broad, such as the widespread physiological changes often referred to as the
"stringent response" mediated by the nucleotides guanosine tetraphosphate (ppGpp)
and pentaphosphate (pppGpp), collectively (p)ppGpp. These molecules reduce rep-
lication, transcription, translation and GTP synthesis by direct inhibitory interactions
with essential enzymes of these processes [1,2]. They also stimulate the expression
of genes encoding proteins mediating stress protection (e.g., *B. subtilis hpf* [3]), but
the mechanistic basis for this stimulation is not well understood.

In both Gram-positive and Gram-negative bacteria, RelA/SpoT Homolog (RSH)
enzymes [4] synthesize (p)ppGpp in response to diminished amino acid availability
as manifested by the presence of uncharged tRNA molecules in ribosome A-site [5].
RSH enzymes contain a synthetase domain that transfers the β- and γ-phosphate
groups from ATP and adds them to the 3' hydroxyl of the ribose of a GTP or GDP
molecule to form pppGpp or ppGpp, respectively [6]. In Gram-positive species,
Small Alarmone Synthetase (SAS) proteins, which have homology to the synthetase
domains of RSH proteins [4], also produce (p)ppGpp [7] in a non-ribosome depen-
dent fashion.

Extensive amino acid contacts with the ribosome mediate the regulation of
RSH-dependent (p)ppGpp synthesis in response to amino acid availability [8–10].
Additionally, the RSH proteins *E. coli* RelA [11] and *B. subtilis* Rel [10] are alloster-
ically activated by pppGpp binding. Allostery is also observed with the *B. subtilis*
SAS SasB, where mutation of a single residue prevents allosteric stimulation *in vitro*
[12,13]. Introduction of this mutation into the chromosomal copy of *sasB* mimics a
Δ*sasB* mutation in the context of the regulation of protein synthesis [13]. However,
the overall role of allostery in the regulation of (p)ppGpp metabolism *in vivo* is not
well characterized.

Many RSH enzymes also exhibit hydrolase activity that converts pppGpp and
ppGpp into GTP and GDP, respectively [6]. By comparison with (p)ppGpp synthe-
sis, regulation of RSH-dependent (p)ppGpp hydrolysis is less well understood. In *E.
coli* the RSH protein SpoT serves as the primary hydrolase and strains expressing
a hydrolysis-deficient *spoT* mutant display increased (p)ppGpp abundance during a
diauxic shift [14]. SpoT hydrolysis activity is regulated by protein-protein interactions
with various proteins [15–17]. Most RSH proteins contain a small molecule binding
domain (ACT) and the binding of branched chain amino acids to ACT can stimulate
(p)ppGpp hydrolysis in *R. capsulatus*, but the generality of this mechanism is not

known [18]. In addition, some bacteria contain so-called Small Alarmone Hydrolase (SAH) enzymes capable of hydrolyz-ing (p)ppGpp [4] but their regulation is not well understood.

RSH enzymes, particularly those of the RelA/SpotT/Rel class, exhibit a functional coupling of synthetase and hydrolytic activities, mediated, at least in part, by pppGpp-dependent allosteric regulation and also fundamentally by the aminoacy-lation state of the tRNA in the ribosome A site. [19–21]. Thus, under conditions where the population of tRNAs is partially aminoacylated [22], RSH enzymes would likely oscillate between synthetase and hydrolase activities resulting in changes in (p)ppGpp abundance during growth. Such a mechanism could underlie the proposed ability of (p)ppGpp to act as a reversible brake on growth [23]. In fact, during strong heat shock, the abundance of (p)ppGpp increases quickly (within 2 min) but is substantially dissipated by 15 min [3]. Existing methods of measuring (p)ppGpp abundance including radio-labeling of cells with $^{32}$P-ATP under low phosphate conditions followed by analysis of cell extracts by thin layer chroma-tography (TLC) or liquid chromatography/mass spectrometry (LC/MS) analysis of lysates (e.g., [24–27]) are relatively cumbersome and are not optimal techniques for conditions such as growth that may require both temporal sensitivity and physiological robustness.

Here, we use a (p)ppGpp-sensitive riboswitch [28] in conjunction with firefly luciferase (luc), an unstable reporter with a half-life in *B. subtilis* of ~5 min [29], to serve as a dynamic sensor of (p)ppGpp abundance. We use this sensor to directly demonstrate the roles of the hydrolytic activity of the RSH protein Rel and allosteric regulatory sites in both Rel and a SAS protein in governing changes in (p)ppGpp abundance during physiological contexts including nutrient exhaustion and amino acid downshifts. In addition, we demonstrate that (p)ppGpp synthesis closely coincides with the expression of genes under (p)ppGpp control, indicating that this regulation is rapid and likely direct.

## Results

We placed a sequence corresponding to the (p)ppGpp-sensitive riboswitch from the promoter of *Desulfitobacterium hafniense ilvE* [28] between an inducible promoter ($P_{hyperspank}$) and the *luc* gene encoding firefly luciferase (luciferin 4-monooxygenase) (RsFluc; Figs 1A and S1). This construct was integrated at the *sacA* locus and luminescence mea-surements were performed during growth in S7/glucose defined medium in a microplate reader. Raw luminescence read-ings were normalized to $OD_{600}$ (RLU/OD) (Fig 1B, blue). We observed two waves of luminescence, one just after initiation of growth and then a second larger one during early transition phase, near the time of departure from the most rapid period of growth. While the time and magnitude of the second wave (~220 min) was robust across numerous biological replicates, the first wave was much more variable, possibly as a result of the proximity to the culture dilution performed at the initiation of the experiment. Both waves were dependent on the presence of the riboswitch in the reporter construct as a reporter (RsFluc$^0$) lacking the intervening riboswitch sequence exhibited greatly reduced luminescence (S2A Fig). The residual response suggests that (p)ppGpp may be also stimulating transcription initiation from $P_{hyperspank}$.

An RsFluc-expressing strain carrying deletions in the gene encoding all three *B. subtilis* (p)ppGpp synthetases (*relA*, *sasA*, *sasB*; (p)ppGpp$^0$) exhibited a significantly attenuated luminescence (Fig 1B, red), consistent with the expectation that the signal reflects (p)ppGpp abundance. In addition, we constructed mutant RsFluc reporters containing two previ-ously characterized riboswitch mutations [28], one that decreases sensitivity to (p)ppGpp as the binding pocket has been disrupted (M9) and a second that strengthens the terminator (M11), thereby increasing the specificity of the response (S1 Fig). Reporters containing either mutation exhibited reduced luminescence (S2B, S2C Fig) and a reporter containing both (RsFluc$^{mut}$) was substantially attenuated (Fig 1B, green). As these mutations are not known to completely prevent (p)ppGpp binding, consistently, this effect is less strong than in the (p)ppGpp$^0$ background, which may exhibit some secondary physiological effects resulting from the absence of the signaling molecule, particularly in transcription and translation [1,2].

As a further confirmation of the reporter, we measured (p)ppGpp by LC/MS at points in the growth curve with differ-ent RsFluc activities. Values were normalized with respect to $OD_{600}$ and to a parallel measurement in a (p)ppGpp$^0$ strain,

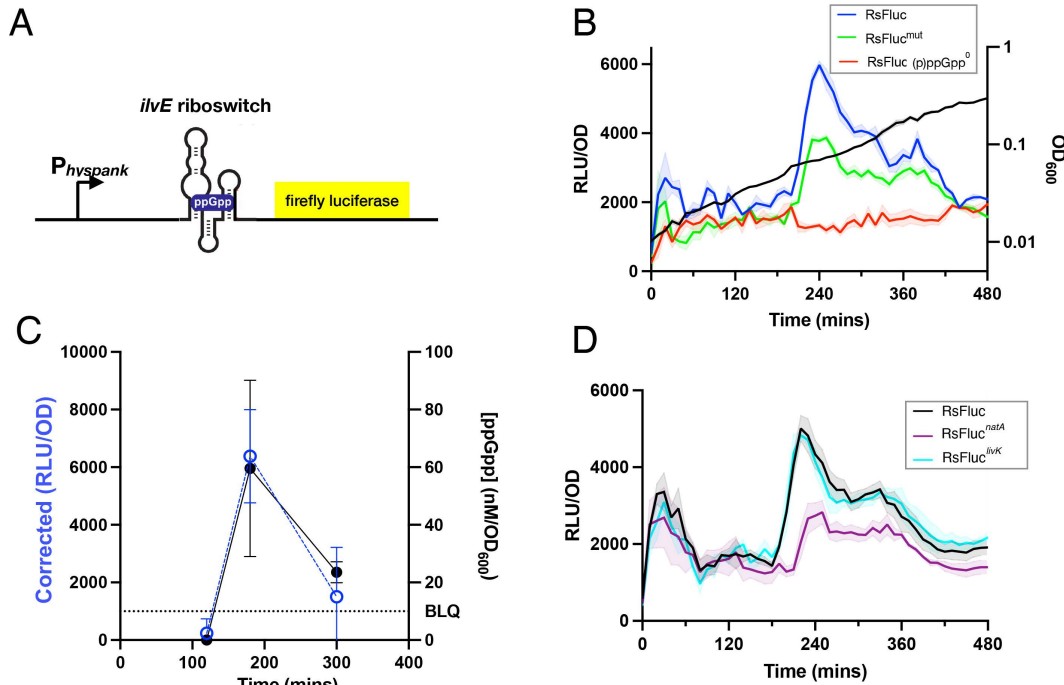

**Fig 1. RsFluc (p)ppGpp-sensitive firefly luciferase reporter.** A, schematic of the (p)ppGpp reporter RsFluc depicting the inducible promoter $P_{hyperspank}$ fused to the (p)ppGpp-senstive riboswitch of *D. hafniense ilvE* followed by the gene encoding firefly luciferase. B, luminescence (RLU/OD$_{600}$) of RsFluc (blue; JDB4496), mutant RsFluc$^{mut}$ (green; JDB4631) in wildtype backgrounds and RsFluc in a Δ*relA*Δ*sasA*Δ*sasB* background (red; JDB4512), C) ppGpp concentration determined by HPLC-MS (black closed circles) and corresponding reporter luminescence corrected for (p)ppGpp$^0$ luminescence by subtraction (blue open circles). D, luminescence per OD$_{600}$ of RsFluc (black), RsFluc$^{natA}$ (purple, JDB4730) and RsFluc$^{livK}$ (cyan, JDB4731). Shown are representative examples of at least three biological replicates, each comprising three technical replicates.

resulting in lower (p)ppGpp before and after the time of the luminescence peak (Fig 1C). Calculation of the absolute concentration of (p)ppGpp yielded an estimate of ~60 nM/OD$_{600}$ (S3A Fig). Of note, we observed only modest declines in GTP (S3B Fig). Additionally, we confirmed that the RsFluc reporter signal is accompanied by changes in transcriptional read-through by fluorescent *in situ* hybridization (FISH) of the mRNA of a reporter gene downstream of the riboswitch. Approximately 10 minutes after the observed increased transcription at TP3 (S4B Fig), the RsFluc signal reaches maximum (S4A Fig). Transcription rates fall to basal levels in the subsequent timepoints (S4B Fig) and RsFluc signal decreases (S4A Fig). Taken together, these results and those described above demonstrate that RsFluc activity reflects changes in (p)ppGpp abundance.

As noted above, (p)ppGpp refers collectively to ppGpp and pppGpp. The *D. hafniense ilvE* riboswitch has no *in vitro* preference for binding pppGpp or ppGpp [30], suggesting that RsFluc is equally sensitive to both molecules *in vivo*. Aptamers with preference for pppGpp or ppGpp have been identified [30]. For example, the *Clostridiales bacterium natA* aptamer exhibits a ~10-fold preference for pppGpp as compared to ppGpp whereas the *Oxobacter pfennigii livK* aptamer exhibits a ~6-fold preference for ppGpp [30]. We constructed versions of the RsFluc reporter containing these aptamers in place of *D. hafniense ilvE*, RsFluc$^{natA}$ and RsFluc$^{livK}$, respectively. The similarity between RsFluc and the ppGpp sensitive reporter RsFluc$^{livK}$ (Fig 1D) is consistent with previous observations that the abundance of ppGpp > pppGpp [31]. In addition, the pppGpp sensitive reporter RsFluc$^{natA}$ exhibited a delayed expression in comparison to the other reporters (Fig 1D).

We then characterized the effect on RsFluc activity of perturbations that stimulate *in vivo* (p)ppGpp synthesis. As uncharged tRNAs activate RelA-dependent (p)ppGpp synthesis [5], many such perturbations affect aminoacyl-tRNA charging. One example is a culture grown in amino acid-replete medium shifted to a medium lacking amino acids, which is a transition that reduces aminoacylation [32] and stimulates (p)ppGpp synthesis [33]. Consistently, a spike in RsFluc activity occurred shortly (~30 min) after the down-shift (Fig 2A, black) that is attenuated in strain expressing the RsFluc$^{mut}$ reporter. The timing of this response illuminates the temporal relationship between our reporter and (p)ppGpp abundance. Amino acid starvation results in rapid diminution of tRNA charging (~5 min, [34]) and increased ppGpp synthesis (~10–15 min [35]). The difference between these values and that which we observe with RsFluc suggests that activation of the riboswitch and subsequent expression of the firefly luciferase together take <20 min.

Inhibitors of tRNA synthetases serve as an alternative strategy to modulate tRNA charging. For example, mupirocin, an inhibitor of isoleucyl-tRNA synthetase [36] stimulates (p)ppGpp synthesis [37]. Consistently, RsFluc-expressing cells treated with mupirocin exhibited a significant increase in luciferase production not observed in cells expressing the (p)ppGpp-insensitive RsFluc$^{mut}$ reporter, indicating that the effect was specific to changes in (p)ppGpp abundance (Fig 2B). We investigated the mupirocin concentration causing the maximal stimulation of RsFluc activity, observing that 100 ng/ml was less effective than 50 ng/ml (S5A Fig). The severe diminution of growth in the presence of 100 ng/ml

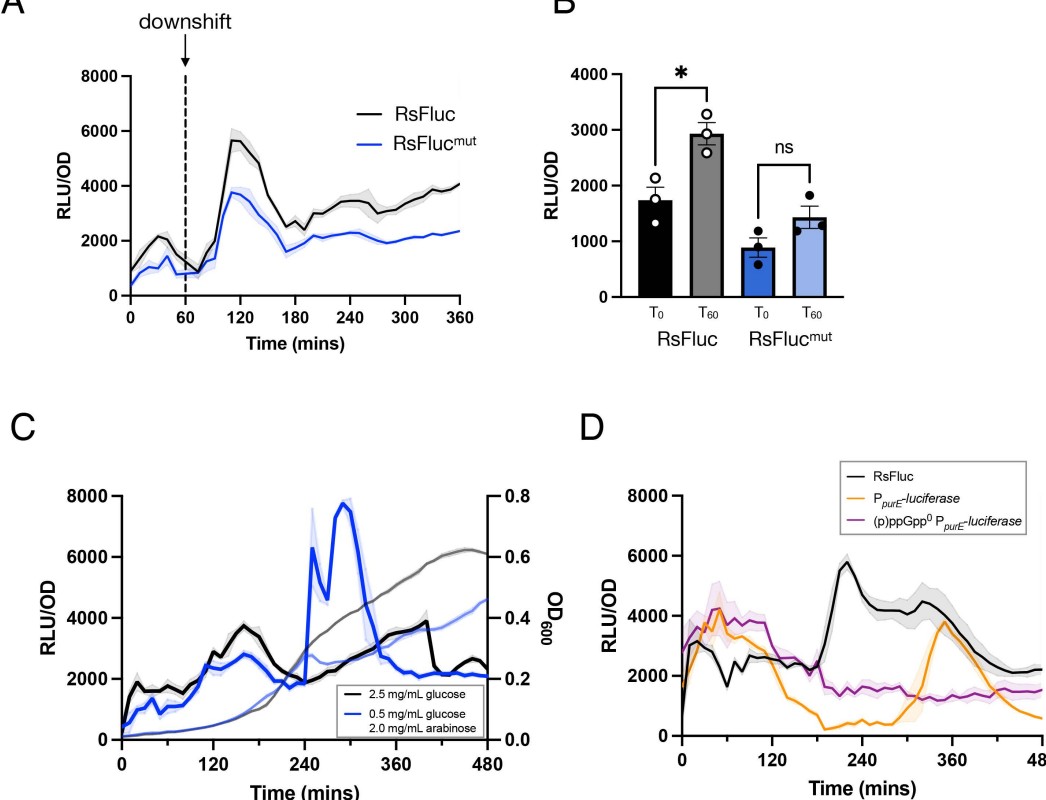

**Fig 2. RsFluc reporter response to stimulation of (p)ppGpp synthesis.** Luminescence (RLU/OD$_{600}$) of: A, RsFluc (black; JDB4496) and RsFluc$^{mut}$ (blue; JDB4631) after amino acid downshift at T$_{60}$ (dashed line); B, RsFluc (black) and RsFluc$^{mut}$ (blue) after 0 and 60 mins of 50 ng/mL mupirocin treatment. Significance was determined by two-tailed t test: p-value <.05 for RsFluc, no significant difference for RsFluc$^{mut}$; C, RsFluc grown in defined minimal media containing either 2.5 mg/mL of glucose (black) or 0.5 mg/mL glucose and 2.0 mg/mL arabinose (blue). Respective OD$_{600}$ measurements are shown in grey and light blue, respectively, and D) RsFluc (black) and a P$_{purE}$ luciferase promoter fusion in wildtype (orange; JDB4803) and (p)ppGpp$^0$ (purple; JDB4804) backgrounds. Shown are representative examples of at least three biological replicates, each comprising three technical replicates.

mupirocin (S5B Fig) suggests at this concentration, the mupirocin is having physiological effects beyond (p)ppGpp synthesis such as direct inhibition of protein synthesis.

Both direct ($^{32}$P-radiolabeling) [14] and indirect (transcriptional profiling) [38] assays observe (p)ppGpp synthesis during a diauxic shift when bacteria adapt to grow on a secondary carbon source following glucose exhaustion. We measured RsFluc activity during a diauxic shift from glucose to arabinose as the only carbon sources (e.g., without amino acid supplementation) following a protocol [29] based on the original Monod *B. subtilis* observations [39]. The transient flattening of the growth curve in the culture grown in 0.5 mg/ml glucose & 2.0 mg/ml arabinose is characteristic of a diauxic shift and is accompanied by prominent spike in RsFluc activity at ~240 min indicative of a substantial increase in (p)ppGpp abundance (Fig 2C, blue). In contrast, growth in 2.5 mg/ml glucose (black) did not result in substantial changes in either growth or luciferase.

As an additional confirmation that RsFluc activity reflects (p)ppGpp abundance, we monitored expression of a firefly luciferase fusion to the *purE* promoter that is sensitive to (p)ppGpp levels [40]. Specifically, (p)ppGpp enhances PurR DNA binding, thereby increasing repression of *purE* transcription. Consistent with previous observations [40], $P_{purE}$-luciferase activity was affected by the absence of (p)ppGpp (Fig 2D, compare orange and purple). We observed lowest $P_{purE}$-luciferase (orange) activity at approximately the same time as RsFluc (black) activity increased (~180 min) and a later rise in $P_{purE}$ activity when RsFluc was diminishing. Taken together, these data support the interpretation that RsFluc activity reflects (p)ppGpp abundance.

We then investigated the source of the (p)ppGpp responsible for the RsFluc signal. *B. subtilis* RelA, SasA (RelP), and SasB (RelQ) are the known (p)ppGpp biosynthetic enzymes [7,41]. Consistently, an RsFluc-expressing strain lacking all three proteins (Δ*relA*Δ*sasA*Δ*sasB*) exhibits substantially reduced luminescence compared to the wild type parent (Fig 1B). We investigated the contributions of each protein to RsFluc activity by assaying cells carrying single mutations in their respective genes under growth (Fig 3A, see S6 Fig for corresponding growth curve) and amino acid downshift (Fig 3B, see S8 Fig for corresponding growth curve). While all three mutations affected expression during growth, a strain lacking RelA synthetase activity due to an inactivating point mutation (D264G) in the synthetase domain [7] exhibited the largest effect (blue, Fig 3A), essentially indistinguishable from the triple mutant Δ*relA*Δ*sasA*Δ*sasB* strain (red). In contrast, single Δ*sasA* (green) or Δ*sasB* (orange) or the combination Δ*sasA*Δ*sasB* (S7 Fig, brown) mutant strains exhibited at most only

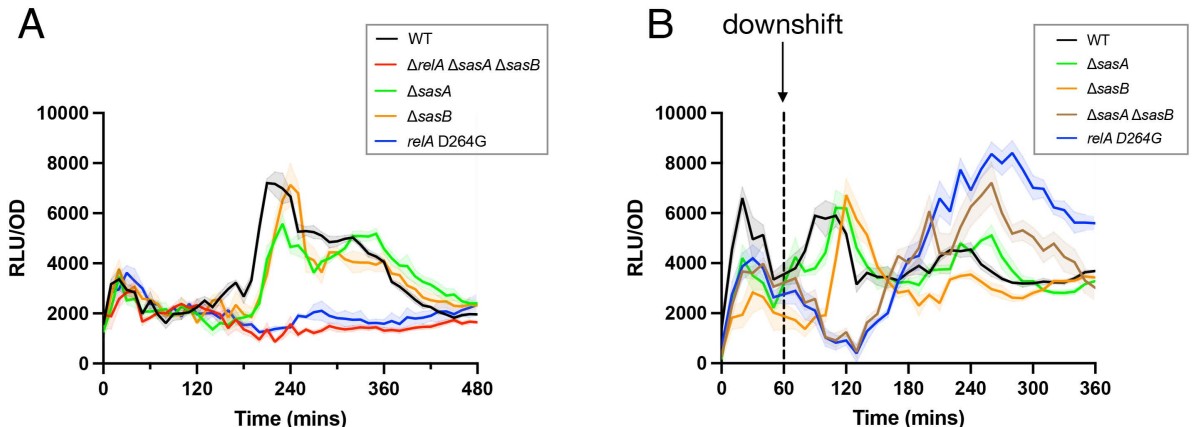

**Fig 3. Relative contributions of (p)ppGpp synthetases to dynamics of RsFluc activity.** Luminescence (RLU/OD$_{600}$) of: A, RsFluc in wildtype (black, JDB4496), Δ*relA*Δ*sasA*Δ*sasB* (red, JDB4512), Δ*sasA* (green, JDB4515), Δ*sasB* (orange, JDB4516), and *relA-D264G* (blue, JDB4741) backgrounds and B), wildtype (black), Δ*sasA* (green), Δ*sasB* (orange), Δ*sasA*Δ*sasB* (brown, JDB4511), and *relA-D264G* (blue) strains after amino acid downshift at T$_{60}$ (dashed line). Shown are representative examples of at least three biological replicates, each comprising three technical replicates.

modest delays in RsFluc signal. During amino acid downshift, the *relA-D264G* strain was again severely affected, but under these conditions, the Δ*sasA*Δ*sasB* strain was similarly defective (blue, brown; Fig 3B).

RsFluc activity in the Δ*sasB* strain was strikingly different as compared to the wildtype parent in the context of nutrient downshift (Fig 3B). What underlies this effect? *In vitro*, SasB is activated allosterically by pppGpp produced by Rel [12]. SasB Phe-42 is a key residue in this regulation and a SasB mutant protein carrying an F42A substitution is no longer allosterically activated *in vitro* by pppGpp [12]. RsFluc activity of a strain carrying a *sasB-F42A* allele in the chromosome [13] was delayed both during growth (Fig 4A, blue) and nutrient downshift (Fig 4B, blue). Thus, this delay is the result of the absence of SasB stimulation. Rel itself is subject to allosteric regulation *in vitro* by pppGpp [10], and Tyr-200 is import-ant for this effect [19]. Introduction of a Rel Y200A mutation into the chromosomal copy of *rel* delayed the increase in RsFluc activity during growth as compared to the wildtype parent strain (Fig 4A, purple). During a nutrient downshift (as in Fig 3B), this mutation also delayed increased RsFluc activity (Fig 4B, purple). Importantly, the magnitude of the increase was not substantially affected, indicating that Rel Y200A likely does not impair synthetase activity during nutrient down-shift. Thus, taken together, these observations provide the first evidence of the *in vivo* function of allosteric regulation in (p)ppGpp metabolism.

Regulation of RSH enzymes is not limited to direct effects on their biosynthetic activity as many RSH enzymes includ-ing Rel also hydrolyze (p)ppGpp, thereby indirectly antagonizing the synthetic activity. Investigating the contribution of Rel-dependent (p)ppGpp hydrolysis to RsFluc expression is complicated by its essentiality in the presence of any (p)ppGpp synthetases, including Rel itself [41]. Inspired by a previous study [3], we developed a system allowing transient expression of a RelA mutant lacking hydrolytic activity (D78A; [21]) under control of the P$_{lial}$ bacitracin-inducible promoter [42] (Fig 5A). We first confirmed that expression of an inducible wildtype *rel* allele complemented RsFluc activity in a ppGpp$^0$ background (compare blue and gray; Fig 5B). Expression of a Rel-D78A mutant protein slowed growth (S11 Fig), consistent with increased (p)ppGpp abundance and produced a broadening of the increased RsFluc activity before and after the peaks seen with expression of wildtype RelA (Fig 5B, red). Another reported (p)ppGpp hydrolase is NahA [43], but a Δ*nahA* mutation does not affect RsFluc (S12 Fig). Thus, Rel is the primary hydrolase activity affecting the dynamics of (p)ppGpp abundance under our experimental conditions.

The importance of Rel (p)ppGpp synthetic activity for RsFluc expression (Fig 4A) suggests that amino acid abundance is the primary factor determining for stimulating (p)ppGpp synthesis. To test this hypothesis, we examined RsFluc activity

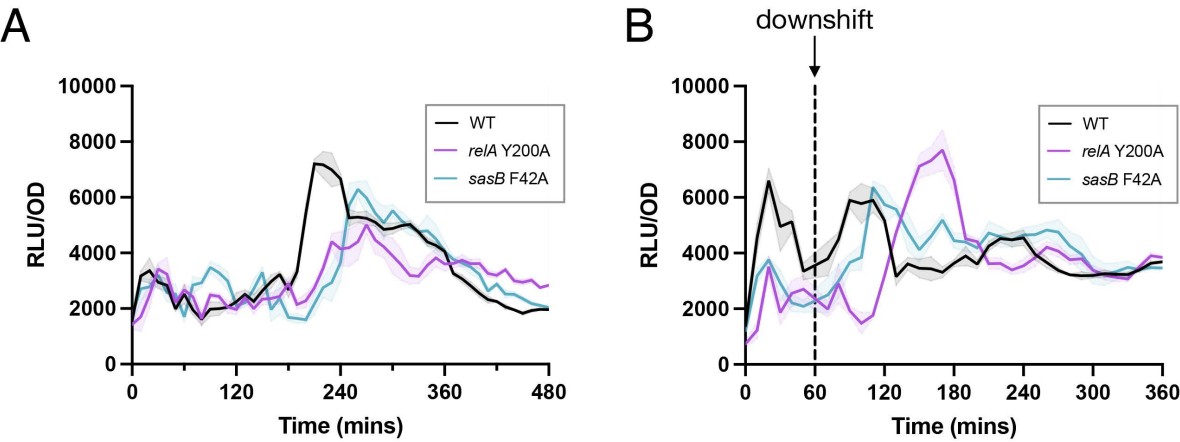

**Fig 4. RsFluc activity coordinated by allosteric network.** Luminescence (RLU/OD$_{600}$) of: A, RsFluc in wildtype (black, JDB4496), *relA-Y200A* (fus-chia, JDB4528), and *sasB-F42A* (blue, JDB4711) strains and B, RsFluc in wildtype (black), *relA-Y200A* (fuschia), and *sasB-F42A* (blue) strains after amino acid downshift (dashed line). Shown are representative examples of at least three biological replicates, each comprising three technical replicates.

PLOS Genetics

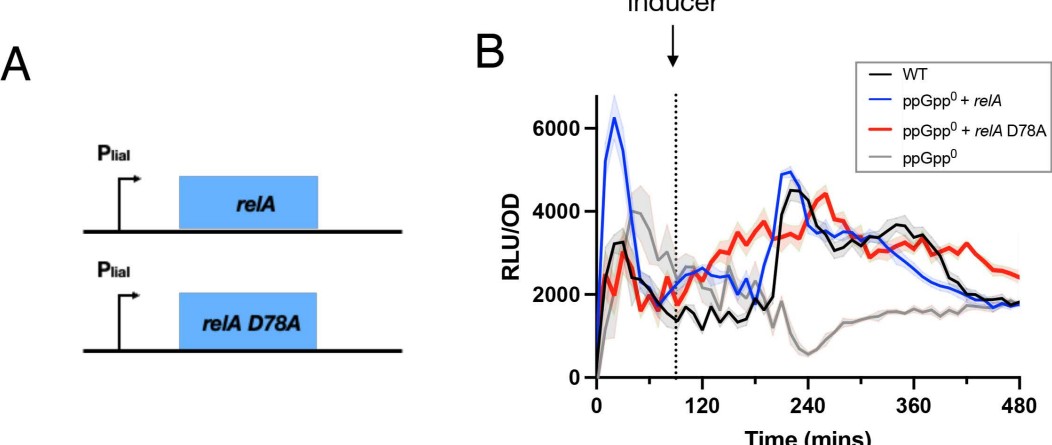

**Fig 5. RelA hydrolase activity contributes to dynamics of RsFluc activity.** A, schematic of bacitracin-inducible *relA* constructs. B) luminescence (RLU/OD$_{600}$) of RsFluc in wildtype (black, JDB4496), Δ*relA*Δ*sasA*Δ*sasB* with P$_{lial}$-*relA* (blue, JDB4675), Δ*relA*Δ*sasA*Δ*sasB* with P$_{lial}$-*relA-D78A* (red, JDB4676) and Δ*relA*Δ*sasA*Δ*sasB* (gray, JDB4512) strains. RelA expression induced by 5 μg/mL bacitracin at T$_{90}$ (dashed line). Shown is a representative example of at least three biological replicates, each comprising three technical replicates.

as a function of amino acid concentration in the growth medium. Consistently, increasing concentrations of Casamino acids (CAA) delay maximal RsFluc activity (Fig 6A) with minimal effects on growth. In the absence of added amino acids, RsFluc activity was substantially attenuated, especially following departure from exponential growth (Fig 6B). This activity is substantially higher than exhibited by the RsFluc$^{mut}$ (p)ppGpp-insensitive reporter, so it likely reflects actual (p)ppGpp abundance.

We finally investigated how changes in RsFluc activity correlate with the appearance of known (p)ppGpp-dependent physiological phenomena in *B. subtilis* including gene activation [44] and protein synthesis attenuation [45]. Specifically, induction of the stringent response by arginine hydroxamate treatment leads to (p)ppGpp-dependent expression of a number of amino acid biosynthetic genes [44]. We constructed firefly luciferase fusions to the promoters of three of these

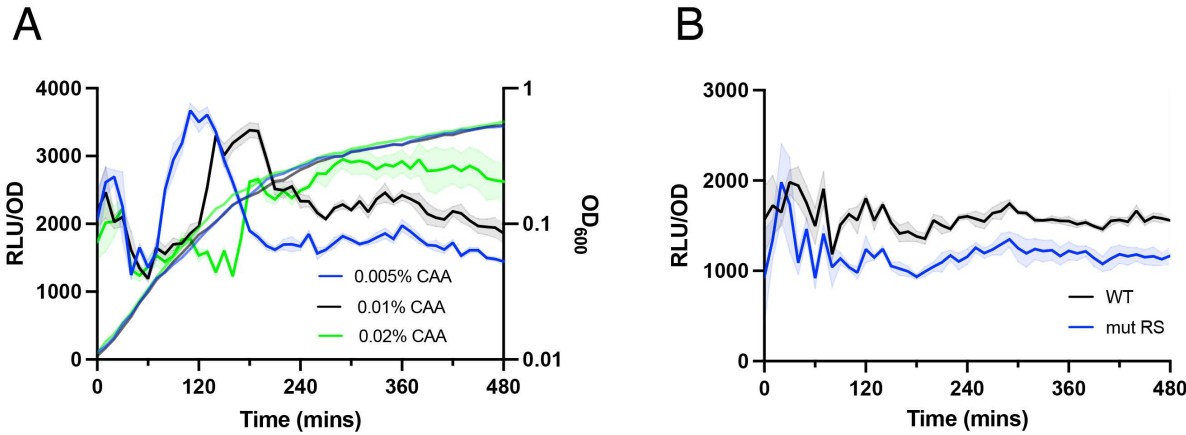

**Fig 6. RsFluc activity and amino acid availability.** Luminescence (RLU/OD$_{600}$) of: A, RsFluc in prototroph (JDB4656) cultured in S7 supplemented with 0.005% (blue), 0.01% (black), and 0.02% (green) casamino acids and B, RsFluc (black) and RsFluc$^{mut}$ (blue) in prototroph background without amino acid supplementation. Shown are representative examples of at least three biological replicates, each comprising three technical replicates.

genes ($P_{serA}$-luc, $P_{ilvB}$-luc, $P_{metE}$-luc) and compared their temporal dynamics with RsFluc. All reporters exhibited luciferase activity with very similar temporal dynamics to that of RsFluc (Fig 7A). Consistent with the previous study, presence of a *relA-D264G* mutation substantially attenuated all reporters (Fig 7B) [44].

The increase in RsFluc coincides with a sharp reduction in growth rate (Fig 8A, black). The linear correlation between protein synthesis (particularly ribosomal proteins) and growth rate [46] suggests that the increase in RsFluc is associated

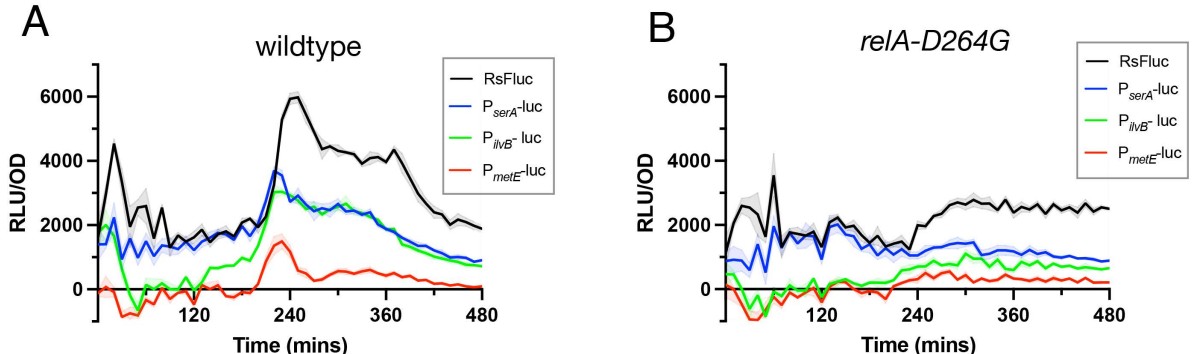

**Fig 7. Correlation of amino acid biosynthetic gene expression and RsFluc activity.** Luminescence (RLU/OD$_{600}$) of: A, RsFluc (black, JDB4496), $P_{serA}$-luc reporter (blue, JDB4759), $P_{ilvB}$-luc reporter (green, JDB4792), and $P_{metE}$-luc reporter (red, JBD4798) in the wildtype background and B, reporters as A in the *relA-D264G* genetic background. Shown are representative examples of at least three biological replicates, each comprising three technical replicates.

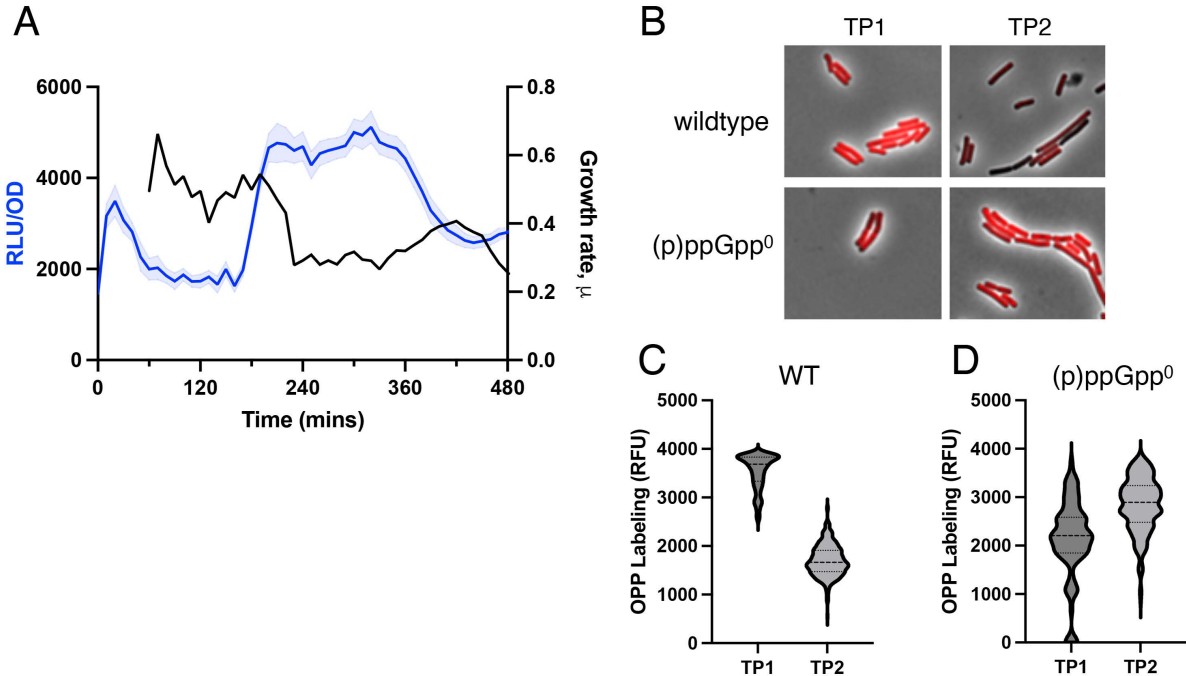

**Fig 8. Temporal relationship between protein synthesis and RsFluc activity.** A, luminescence (RLU/OD$_{600}$) of RsFluc (blue, JDB4496) and growth rate (black). Growth rate/hour ($\mu$) at a given time (t) is defined as $\log_2[OD_{600}(t)-OD_{600}(t_{-60})]$. B, representative composite (fluorescent, phase) images of cells at TP1 and TP2 (see S11 Fig) in either wildtype (JDB4486) or $\Delta relA\Delta sasA\Delta sasB$ ('(p)ppGpp$^0$'; JDB4512) backgrounds. C) and D) quantification of images from B.

PLOS Genetics

with reduced protein synthesis. To compare the temporal dynamics of protein synthesis and (p)ppGpp, we incorporated OPP (O-propargyl-puromycin) into cells before (S13 Fig, arrow "TP1") and during ("TP2") the increase in RsFluc. Click-conjugation of OPP with a fluorophore (Fig 8B, 'wildtype') and subsequent quantification revealed OPP labeling substantially declined in the interval from TP1 to TP2 (Fig 8C). Since (p)ppGpp inhibits translation initiation [45], we asked if this change in OPP labeling was dependent on (p)ppGpp. We obtained samples of ppGpp$^0$ cells at the same time points and processed them similarly (Fig 8B, 'ppGpp$^0$'). In contrast to the wildtype cells, OPP labeling of ppGpp$^0$ cells did not decline between TP1 and TP2, and in fact, it increased (Fig 8D), consistent with the role of (p)ppGpp in attenuating protein synthesis.

## Discussion

Here we develop and evaluate RsFluc, a novel dynamic riboswitch-based reporter of (p)ppGpp metabolism in *B. subtilis*. We confirm that RsFluc accurately reflects (p)ppGpp abundance using a variety of approaches, including mutagenesis of both reporter and host. In addition, known stringent response inducers such as inhibition of tRNA aminoacylation or a nutrient downshift robustly stimulate RsFluc activity. We use RsFluc to demonstrate the *in vivo* relevance of mutations that affect allosteric activation of (p)ppGpp synthetases and the involvement of Rel hydrolase activity in (p)ppGpp abundance. Together, these observations demonstrate that RsFluc will serve as an integral component of the experimental toolkit in investigating (p)ppGpp metabolism and regulation. Finally, we demonstrate the temporal dynamics of RsFluc activity correlate with the appearance of known (p)ppGpp-dependent physiological phenomena in *B. subtilis* including gene activation and protein synthesis attenuation, suggesting that these effects are likely direct.

The importance of Rel, the sensor of amino acid availability (Fig 3A), and the effect of supplemental amino acid concentrations for RsFluc activity (Fig 6A) are consistent with the long-held view that amino acid limitation is the key signal underlying (p)ppGpp synthesis. To further investigate this, we grew a prototrophic strain in the absence of supplemental amino acids where amino acid abundance is limited only by the availability of carbon and nitrogen. We observed no substantial fluctuations in RsFluc activity during growth (Fig 6B) and reduced levels throughout growth. Importantly, the basal level of RsFluc activity is higher than that of RsFluc$^{mut}$ (compare black and blue), indicating that these levels, even if low, are real. This data may be relevant to understanding the role of (p)ppGpp role in contexts sometimes referred to "basal" levels of (p)ppGpp, which are defined as non-stressful growth conditions [23,47,48] such as motility during exponential growth [49] and the coordination of ribosome synthesis to functional tRNA abundance [22,50].

Investigation of (p)ppGpp metabolism in both *E. coli* [51] and *B. subtilis* [52] observed a decrease in GTP abundance accompanying an increase of (p)ppGpp abundance consistent with GTP/GDP serving as the biosynthetic precursor. However, similar to *E. coli* during nutrient limitation conditions [24] or *B. subtilis* during heat shock [3], we do not observe a similar (negative) correlation (S3B Fig). This discrepancy could be a consequence of differences in the kinetics of activation, as the hydrirocin used in the earlier studies rapidly and strongly stimulates (p)ppGpp abundance in <5min. Thus, rapid synthesis of (p)ppGpp could deplete GTP/GDP pools before homeostatic mechanisms [53] are able to restore their levels. Alternatively, this discrepancy may reflect the magnitude of changes in tRNA charging: under explicit amino acid starvation, tRNA charging falls to near-zero levels on a time scale of ~min [34], a much more substantial decrease than observed during growth phase transitions [22].

(p)ppGpp has well-characterized global effects [1,2] that could affect RsFluc expression, apart from the riboswitch element, and complicate interpretation of RsFluc activity. However, we believe this is unlikely for two reasons. First, activity of the RsFluc$^{mut}$ reporter, carrying a mutation in the *ilvE* riboswitch that reduces the *in vitro* sensitivity to (p)ppGpp [28] is attenuated as compared to the wildtype RsFluc reporter (Fig 1B). This attenuation is similar to that observed with mutant RsFluc reporter lacking the entire RS element (RS$^0$, S2A Fig). Thus, these observations in a wildtype background indicate that a significant effect on luciferase activity can be observed even in a strain wildtype for (p)ppGpp synthesis. Further, these observations are consistent with a stimulatory effect of (p)ppGpp on transcription as we observed (Fig 7). Second,

while (p)ppGpp has a clear inhibitory effect on global protein synthesis [45], this effect would tend to dampen the response of a protein reporter such as luciferase thereby minimizing observed (p)ppGpp-dependent increase in RsFluc activity. Thus, this increase (e.g., Fig 1B) is likely less than is actual given the compensatory inhibition of (p)ppGpp on synthesis of luciferase.

Our work confirms the previously reported [7] importance of Rel for (p)ppGpp synthesis during growth (blue; Fig 3A). However, SasA and SasB only modestly contribute to the RsFluc dynamics during transition phase (orange, green; Fig 3A). The absence of a substantial effect of the lack of SasB is surprising given that it is expressed just prior to transition phase and is also a potent producer of ppGpp, at least *in vitro* or when overexpressed ectopically in *E. coli* [7]. The lack of an effect of SasA is also surprising given that both SasA is expressed during the transition from exponential growth [54] like RsFluc and that SasA synthesizes ppGpp [27]. NahA may degrade a substantial fraction of the ppGpp to pGpp [43] but the absence of NahA does not have a substantial effect on RsFluc (S12 Fig). To our knowledge, previous studies have not examined the specific synthetase and hydrolase dependency of (p)ppGpp accumulation during *B. subtilis* growth.

In contrast to growth, *sasA* and *sasB* have substantial effects during amino acid downshift (Fig 3B, green, orange), with strains carrying Δ*sasA* or Δ*sasB* mutations exhibiting delayed RsFluc expression. In fact, the double Δ*sasA*Δ*sasB* mutant strain (brown) exhibits greatly attenuated RsFluc activity, similar to the *relA-D264G* strain (blue). Both strains also exhibit severe growth defects following downshift (S6 Fig, blue, brown), suggesting that all three genes play a key role in the resumption of growth following abrupt starvation, as has been suggested for *E. coli* RelA [55]. Thus, taken together, our observations provide a starting point for future investigations to understand the regulation of *B. subtilis* Rel, SasA, and SasB during regrowth.

The role of (p)ppGpp in the phenomenon of diauxie has been studied for >50 years [56]. However, *E. coli* strains lacking (p)ppGpp exhibit multiple amino acid auxotrophies [57], complicating analysis in a growth medium consisting only of a single carbon source (that is, without added amino acids that can serve as a source of carbon). Similar auxotrophies also have been reported for *B. subtilis* [38,44]. The experiments presented here demonstrate that (p)ppGpp is synthesized during a diauxic shift in *B. subtilis* (Fig 2C). Rel is the only (p)ppGpp synthetase necessary for this synthesis as the RsFluc activity of a Δ*sasA*Δ*sasB* strain is essentially the same as the wildtype parent (S7 Fig). Thus, during glucose exhaustion, the availability of amino acids necessary for charging tRNAs as sensed by Rel is the signal stimulating (p)ppGpp synthesis. Interestingly, *Lactococcus lactis* exhibits single cell heterogeneity in their ability to react to diauxie [58]. Given that this phenomenon is subject to stringent response control, heterogeneity in (p)ppGpp abundance at the single cell level was proposed to underlie this phenomenon. This intriguing hypothesis awaits the development of (p)ppGpp reporters amenable to single cell analysis.

Allosteric regulation has been observed *in vitro* for *B. subtilis* (p)ppGpp synthetases Rel [10] and SasB [12] (but not SasA [59]). Our analysis of strains carrying mutations in residues identified *in vitro* to be important for this regulation (RelA-Y200A [10], SasB-F42A [12]) reveals that that allostery affects (p)ppGpp abundance *in vivo* (Fig 4). In addition, while these strains grow equivalently to the wildtype parent (S9 Fig), both exhibit impaired recovery from nutritional downshift (S10 Fig). This is the first time that allostery has been demonstrated to mediate a specific physiological function of the (p)ppGpp metabolic network. Since strains carrying a Δ*sasB* deletion also exhibit a substantial phenotype with respect to (p)ppGpp synthesis in response to a nutritional downshift (Fig 3B), the phenotype of a *sasB-F42A* strain with respect to recovery suggests a mechanistic explanation for this phenotype.

The synthetic activity of *B. subtilis* (p)ppGpp synthetases is well understood to be regulated by interactions with the ribosome (e.g., Rel [21]) or by transcriptional regulation in the case of SasA/RelP and SasB/RelQ [54]). In contrast, the regulation of hydrolase activity, either in the context of the Rel dual synthetase/hydrolase or standalone hydrolases such as NahA is less clear. A difficulty in investigating the in *vivo* function of Rel hydrolase activity results from its essentiality in the presence of any (p)ppGpp synthetic activity. Our experiments using comparison of a transiently expressed wildtype Rel with a hydrolase-dead mutant (D78A) directly demonstrate that the hydrolase activity of Rel affects the abundance

of (p)ppGpp as verified by the RsFluc reporter (Fig 5B). The relative constancy of luciferase expression in the *relA-D78A* strain as compared with the strain expressing a hydrolase-active allele suggests both that regulation of hydrolysis activity is important in generating the observed spike in ppGpp and that it is not just regulation of the synthetase domain which is important for determining (p)ppGpp abundance. This interpretation is consistent with a recent model of Rel function that envisions a switch between Hydrolase-Active/Synthetase-Inactive and Hydrolase-Inactive/Synthetase-Active configurations [10,21]. This switch could be controlled by ribosome association, but it is not clear what would stimulate a change in this interaction under the gradual nutrient exhaustion investigated here. Alternatively, this could be a consequence of the allosteric regulation of Rel by one of its products [19], as discussed above.

Treatment with a small molecule that reduces tRNA aminoacylation, such as mupirocin, a isoleucyl-tRNA synthetase inhibitor, is a typical laboratory strategy to induce (p)ppGpp synthesis [36]. We observe that mupirocin stimulates RsFluc expression (Fig 2B) but the response is not monotonic, with 50 ng/ml mupirocin stimulating to a greater extent than 25 ng/ml but less than 100 ng/ml (S5A Fig). Given that increased mupirocin results in decreased tRNA charging, which itself inhibits protein synthesis, 100 ng/ml mupirocin could affect protein synthesis sufficiently to interfere with expression of the luciferase reporter. The observation that 100 ng/ml mupirocin affects growth much more substantially than 50 ng/ml (S5B Fig) is consistent with this explanation. Thus, interpretation of the effect of 100 ng/ml mupirocin is complicated by the action of ppGpp to inhibit protein synthesis [45]. Similar issues may be common with other molecules that stimulate (p)ppGpp synthesis by affecting tRNA charging, suggesting that careful titration may be necessary in order to properly control for indirect effects on protein synthesis [60].

The close temporal correlation between the activity of luciferase fusions to the promoters of several amino acid biosynthetic genes with RsFluc (Fig 7A) suggests that (p)ppGpp directly activates their transcription. A fall in GTP levels as a consequence of (p)ppGpp synthesis and the subsequent activation of CodY, a known repressor of amino acid biosynthetic genes is a potential mechanism [44]. However, GTP levels do not significantly fall during the time of RsFluc activity (S2B Fig), similar to that observed when (p)ppGpp and GTP abundances were monitored by HPLC during *E. coli* growth [24]. Finally, (p)ppGpp binds RNAP and in conjunction with DksA directly modulates transcription of target genes in *E. coli* [61], but as there is no evidence for a similar interaction in *B. subtilis*, future research should investigate how such positive regulation occurs.

The use of riboswitches as a basis for physiological reporters of signaling nucleotide abundance is becoming increasingly widespread. Examples include a *B. subtilis* cyclic-di-GMP reporter [62], a *S. aureus* cyclic-di-AMP reporter [63], an *E. coli* (p)ppGpp reporter [64]. These studies utilized a fluorescent reporter and observed changes in fluorescence during growth, although the relative stability of the reporters impaired analysis of dynamics. These metabolites, especially (p)ppGpp, play important roles in the physiology of commensal and pathogenic bacteria. For example, bacteria lacking (p)ppGpp exhibit defects in virulence (e.g., *Campylobacter jejuni* [65]), *Mycobacterium tuberculosis* [66,67]) and commensal persistence [68] and colonization [69]. One issue in investigating these phenomena is that assessment of (p)ppGpp abundance in situations outside of carefully controlled physiological contexts is technically challenging. However, the ability to measure luciferase produced by bacteria during murine infection [70], suggests that it may be possible to monitor expression of the RsFluc reporter inside of a host. In addition, this reporter should facilitate the analysis of (p)ppGpp metabolism in natural isolates and in antibiotic-tolerant persister cells which display intriguing variability in (p)ppGpp abundance [71].

Amino acid availability plays a central role in host-microbe and microbe-microbe interactions. For example, the branched chain amino acids leucine, isoleucine, and valine are essential mammalian amino acids and as such, gut bacteria are a critical source of these molecules [72]. In addition, extracellular amino acid availability/exchange plays a key role in the organization of bacterial communities [73]. Intracellular amino acid availability affects extracellular amino acid availability, either as a consequence of passive and/or active transport across the membrane [74] or potentially as a result of phage lysis [75]. Thus, since (p)ppGpp plays a central role in coordinating intracellular amino acid biosynthesis, (p)ppGpp metabolism likely affects these processes. The RsFluc reporter described here should greatly facilitate investigation of this regulation in physiologically complex contexts.

## Methods

### Strain construction

Strains were derived from *B. subtilis* 168 *trpC2* except as noted and are listed in S1 Table. Strains were constructed by transformation using conventional methodology and where necessary, media was supplemented with either 100 µg/mL spectinomycin, 10 µg/mL kanamycin, 5 µg/mL chloramphenicol, or 1X MLS. Reporters at *sacA* were constructed using pSac-cm [76] derived plasmids (S2 Table), confirmed via whole plasmid sequencing via Plasmidsaurus or Genewiz, and *sacA* integration was confirmed by assaying growth on TSS/glc and nongrowth on TSS sucrose plates. Constructs at *amyE* were constructed using pDG1730 derived plasmids, confirmed via whole plasmid sequencing, and *amyE* integration was confirmed by assaying on LB starch plates. Bacitracin-inducible constructs were designed as described previously [77], using $P_{liaI}$ promoter upstream of induced gene (S2 Table). Marker free site-directed mutagenesis of the *B. subtilis* chromosome was done using pMINIMad2 methodology as described [78], confirming via whole plasmid sequencing and mutated locus sequence by amplification and Sanger sequencing (Genewiz).

### (p)ppGpp reporter construction

The various (p)ppGpp reporter strains were constructed by fusing their respective gBlocks (S3 Table) downstream of IPTG-inducible promoter, $P_{hyperspank}$, and upstream of firefly luciferase gene, using a pSac-cm cloning vector via conventional restriction enzyme and Gibson assembly cloning techniques. Plasmids were sequenced verified and are listed in S2 Table. The vectors were transformed into *B. subtilis*, selecting on LB Agar supplemented with 5 µg/mL chloramphenicol, and *sacA* integration was verified using TSS/glc and TSS sucrose plating.

### Luminescence measurements

Luminescence was measured in a Tecan Infinite M200 Pro instrument with continuous shaking at 37°C, taking $OD_{600}$ and luminescence reads every 10 minutes. Cultures were grown from single colonies grown overnight on LB plates at 37°C. Single colonies were picked into 2 mL S7 + CAA (1X MOPS (Teknova), 1.32 mM $K_2HPO_4$, 1% glucose, 0.1% glutamic acid, 0.01% casamino acids, 40 µg/mL L-Trp), unless otherwise stated, and grown at 37°C in a roller drum for ~3.5 hours. $OD_{600}$ measurements were taken to ensure colonies remained in early exponential phase (between $OD_{600}$ 0.3-0.6). The cultures were then diluted to $OD_{600}$ = 0.05 in 0.5 mL fresh S7 + CAA supplemented with 4.7 mM D-luciferin (Goldbio) and 10 µM IPTG (Goldbio). Note that a spectrophotometer $OD_{600}$ reading of 0.05 is equivalent to a plate reader reading of ~0.01. Cultures were aliquoted in triplicate in wells amounting 150 µL each in a 96-well white-walled, flat and clear bottom plate (Greiner Bio-One). Media only cells were used for background subtraction.

### HPLC-MS nucleotide quantification

Cultures were grown from single colonies grown overnight on LB plates at 37°C. Single colonies were picked into 2 mL S7 + CAA and grown at 37°C in a roller drum for ~3.5 hours. $OD_{600}$ measurements were taken to ensure colonies remained in early exponential phase (between $OD_{600}$ 0.3-0.6). The cultures were back diluted to $OD_{600}$ 0.05 in 20 mL S7 + CAA supplemented with 4.7mM D-luciferin and 10µM IPTG in baffled flasks. The cultures were incubated at 37°C with shaking in a water bath. At 120, 180, and 300 min, 450 µL were sampled for measuring luminescence and $OD_{600}$ via Tecan Infinite M200 Pro plate reader, aliquoting 150 µL in triplicate wells. Media only wells were used for background subtraction. Simultaneously, 5 mL of culture were concentrated on 0.2 µm pore (d = 0.47 mm) cellulose acetate membrane filter (Sartorius) via vacuum filtration. The filter was immediately washed with 1 mL ice cold 1M acetic acid solution containing 1 µg/mL $^{15}$N-ATP (Sigma) and 1 µg/mL $^{15}$N-GTP (Sigma) in a 50 mL conical tube, repeatedly washing the solution over the filter surface using the force of a pipette. Conical tubes were vortexed for 3–5 seconds, and the solution was transferred to 2 mL cryo-vials. Solutions were stored at -80°C. Cold acid nucleotide extractions were performed by thawing samples on ice for

60 minutes, vortexing occasionally. Samples were then re-frozen using liquid nitrogen and lyophilized for 6 hours using a VirTis Benchtop Freeze Dryer. Lyophilized samples were dissolved in 200 µl ice-cold HPLC-grade water, centrifuged at maximum speed for 30 minutes, and clear supernatant was collected for quantification via UHPLC-MS/MS.

UHPLC-MS/MS was performed using an ACQUITY Premier UPLC System coupled with a Waters XEVO TQ-S triple quadrupole mass spectrometer. UPLC was performed on a Hypercarb 2.1x50 mm porous graphitic carbon column (3 µm particle size) using a 10–90% linear gradient of solvent B (0.1% ammonium hydroxide in acetonitrile) in solvent A (0.1 ammonium acetate in water, adjusted to pH 9.5 with ammonium hydroxide) within 10 minutes and a flow rate of 0.3 ml/min. MS/MS analysis was operated in negative ionization mode and a multiple reaction monitoring (MRM) mode was adopted. MassLynx was used to quantify peak intensities.

### Amino acid downshift

After 60 min growth in the plate reader, cultures were collected in sterile microcentrifuge tubes, pelleted by centrifugation, and resuspended in equal volume of S7 lacking CAA. Cultures were returned to the plate reader, taking $OD_{600}$ and luminescence reads every 10 minutes for remainder of assay.

### Mupirocin treatment

After 90 min growth in plate reader, mupirocin was added to the cultures at noted concentrations, and $OD_{600}$ and luminescence measurements were resumed for remainder of assay.

### Diauxic shift

Cultures were grown in S7 + CAA (1X MOPS (Teknova), 1.32 mM $K_2HPO_4$, 0.1% glutamic acid, 0.01% casamino acids, 40 µg/mL L-Trp) with either 2.5 mg/ml glucose or a mixture of 0.5 mg/mL glucose and 2.0 mg/mL arabinose.

### OPP Labeling

Click-iT Plus OPP Protein Synthesis Assay Kit (Invitrogen) was used to label cells with OPP following manufacturer's instructions. 1000 µL of cells at given time points were transferred to disposable glass tubes. OPP was added to a final concentration of 10 µM. OPP incorporation was performed at 37 °C on a roller drum for 20 min and all subsequent steps were done at RT. Cells were fixed by adding formaldehyde to a final concentration of 1%. Cells were fixed for 30 min, harvested by centrifugation at 15k x $g$ for 3 mins, and permeabilized using 100 µL of 0.5% Triton X-100 in PBS for 15 min. Cells were labelled using 100 µL of 1X Click-iT cocktail for 20 min in the dark. Cells were harvested and washed one time using Click-iT rinse buffer and then re-suspended in 20–100 µL of PBS for imaging.

### Fluorescence *in situ* hybridization (FISH)

*yfp* FISH probes (S3 Table) were designed and synthesized with CAL Fluor Red 590 Dye (LGC Biosearch Technologies). FISH imaging was performed as described [79] with minor modifications. Briefly, 2 ml cells from mid or late-log cultures grown in S7 + CAA media were collected and fixed with 1% formaldehyde (final concentration) at room temperature for 30 min. The cells were then harvested by centrifugation at 6000 RPM for 3 min at room temperature and washed with 0.02% saline sodium citrate (SSC, Invitrogen). The cell pellets were resuspended in 300 µl MAAM mix (4:1 V:V dilution of methanol to glacial acetic acid) and incubated at -20 °C for 15 minutes, followed by 1X PBS (from 10X PBS, Invitrogen) wash to remove traces of MAAM. Cells were permeabilized in 200µl PBS containing 350 U µl-1 of lysozyme (Epicenter ready-lyse) for 30 min at 37°C. After permeabilization, cells were washed once with 500 µl PBS. The cells were resuspended in 100 µl Stellaris RNA FISH Hybridization Buffer containing 10% formamide and 12.5 µM reconstituted *yfp* oligo probes for hybridization. The cell-probe mix was incubated in a 30°C water bath overnight. Cells were harvested and

washed with 500 µl of reconstituted Stellaris RNA FISH Wash Buffer A containing 10% formamide and incubated at 30°C water bath for 1 hour. The washing step was repeated one more time to remove the excess probe. The cells were harvested and resuspended in 0.5 ml Stellaris RNA FISH Wash Buffer B and incubated for 5 minutes at room temperature. Finally, the cells were harvested and resuspended in 20–40 µl of PBS with 1µl SlowFade Gold Antifade Mountant (Invitrogen). A non-fluorescent control strain, treated with a *yfp* probe, was used as a control to subtract background and autofluorescence in each channel. Phase contrast and fluorescence images (mCherry (ET Sputter Ex560/40 Dm585 Em630/75)) of bacterial cells immobilized on agarose pads were acquired.

## Microscopy

Microscopy was performed on cells immobilized on 1% agarose pads prepared with PBS. Imaging was performed using a Nikon 90i microscope with a Phase contrast objective (CFI Plan Apo Lambda DM × 100 Oil, NA 1.45), an X-Cite light source, a Hamamatsu Orca ER-AG, and the following filter cube: mCherry (ET Sputter Ex560/40 Dm585 Em630/75). Phase contrast and fluorescence images of bacterial cells immobilized on agarose pads were acquired. The image stacks were analyzed in the software Fiji with the help of the MicrobeJ plugin [80]. The straighten and intensity options in the MicrobeJ plugin were used to measure the average fluorescence per pixel within each cell. A non-fluorescent control strain was used to subtract background and autofluorescence in each channel.

## Supporting information

**S1 Fig.  The sequence upstream of luciferase in RSFluc.** Shown is the $P_{hyperspank}$ promoter sequence (red), the *lac* operator binding sequence (blue) and the *D. hafniense ilvE* riboswitch gBlock sequence (green), with anti-terminator and terminator sequences bolded and underlined, and the M9 and M11 mutations from Sherlock et al 2018 annotated. In gray, the Shine-Delgarno ribosome binding sequence (RBS). In orange, the 5' end of the firefly luciferase gene. Restriction enzymes HindIII and BamHI sites bolded.
(TIFF)

**S2 Fig.  Luminescence activity of RSFluc reporter mutants.** Luminescence ($RLU/OD_{600}$) of: A, RsFluc (blue) and mutant RsFluc construct lacking an aptamer sequence (orange, $RS^0$), B, RsFluc (blue) and M11 mutant RsFluc (green); and C, RsFluc (blue) and M9 mutant RsFluc (pink). Growth ($OD_{600}$) (black).
(TIFF)

**S3 Fig.  HPLC-MS quantifications of nucleotide levels sampled during growth.** Nucleotide quantifications in the WT (black, JDB4496) and a (p)ppGpp$^0$ background (red, JDB4512) of A) ppGpp, B) GTP, C) guanosine pools, and D) ATP, as analyzed via LC-MS collected at specified intervals.
(TIFF)

**S4 Fig.  RSFluc transcriptional activity assayed by FISH.** A) Samples of a strain carrying both RSFluc and *ilvE-riboswitch*-YFP reporters (JDB4623) were collected and analyzed via FISH at specified time points (TP1-TP5: TP1 = 80, TP2 = 120, TP3 = 200, TP4 = 220, TP5 = 240 mins) throughout the luminescence curve (RLU/OD) and imaged via fluorescent microscopy. B) The population distribution of single cell fluorescence at each time point (TP1, n = 202; TP2, n = 333; TP3, n = 481; TP4, n = 315; TP5, n = 197).
(TIFF)

**S5 Fig.  RsFluc response to mupirocin titration.** A, luminescence ($RLU/OD_{600}$) of RsFluc (JDB4496) at start ($T_0$) of treatment (black) and after 60 mins of treatment, $T_{60}$ (blue) with varying concentrations of mupirocin. Significance determined by two-way ANOVA with multiple comparison, comparing $T_0$ and $T_{60}$ under each treatment. 50 ng/mL mupirocin

(*) had a p-value of 0.0036 whereas other treatments were not significant. B, growth (OD$_{600}$) of RsFluc before and after the time of mupirocin addition (dotted line) as treated with DMSO (black), 10 µg/mL (blue), 25 µg/mL (brown), 50 µg/mL (cyan), and 100 µg/mL (green) mupirocin.
(TIFF)

**S6 Fig. Growth curves of (p)ppGpp synthetase mutants.** Growth (OD$_{600}$) of wildtype (black, JDB4496), (p)ppGpp$^0$ (red, JDB4512), Δ*sasA* (green, JDB4515), Δ*sasB* (orange, JDB4516), and *relA-D264G* (blue, JDB4741) strains.
(TIFF)

**S7 Fig. Effect of SasA/B proteins on RsFluc activity.** Luminescence (RLU/OD$_{600}$) of RsFluc in WT (black, JDB4496) and Δ*sasA*Δ*sasB* (gold, JDB4508) backgrounds.
(TIFF)

**S8 Fig. Synthetases necessary for growth recovery from nutrient downshift.** Luminescence (RLU/OD$_{600}$) of RsFluc measured post nutrient downshift at T$_{60}$ (dashed line) in WT (black, JDB4496), Δ*sasA* (green, JDB4515), Δ*sasB* (orange, JDB4516), Δ*sasA*Δ*sasB* (gold, JDB4508), and *relA-D264G* (red, JDB4741) backgrounds.
(TIFF)

**S9 Fig. Growth of (p)ppGpp allosteric synthetase mutants.** Growth (OD$_{600}$) of wildtype (black, JDB4496), *relA-Y200A* (fuschia, JDB4528), and *sasB-F42A* (sky blue, JDB4711) strains.
(TIFF)

**S10 Fig. Allosteric regulation affects recovery post downshift.** Growth (OD$_{600}$) post nutrient downshift at T$_{60}$, (dashed line) in wildtype (black, JDB4496), *relA-Y200A* (fuschia, JDB4528), and *sasB-F42A* (sky blue, JDB4711) strains.
(TIFF)

**S11 Fig. Growth of inducible WT and RelA-D78A.** Growth (OD$_{600}$) before and after 5 µg/mL bacitracin addition (dashed line) in WT (black, JDB4496), (p)ppGpp$^0$ with inducible WT *relA* (blue, JDB4675), and (p)ppGpp$^0$ with inducible *relA-D78A* (green, JDB4676) backgrounds.
(TIFF)

**S12 Fig. NahA contribution to RsFluc activity.** Luminescence (RLU/OD$_{600}$) of RsFluc in wildtype (black, JDB4496) and Δ*nahA* (purple, JDB4567) backgrounds.
(TIFF)

**S13 Fig. Time points for OPP labeling.** Luminescence (RLU/OD$_{600}$) of RsFluc in wildtype (blue, JDB4496). Samples were taken at time points TP1 (120 min) and TP2 (240 min).
(TIFF)

**S1 Table. Strains used.**
(DOCX)

**S2 Table. Plasmids used.**
(DOCX)

**S3 Table. Oligonucleotides used.**
(DOCX)

**S1 Data. Sheet "Figs 1B, S2A, S2B, S2C".** Raw luminescence data relevant to Figs 1B and 2A–2C.Sheet "Figs 1C, S3A, S3C, S3D". Raw mass spectrometry data relevant to Figs 1C and S3A–S3C. Sheet "Fig 1C". Raw luminescence

data relevant to Fig 1C. Sheet "Fig 1D". Raw luminescence data relevant to Fig 1C. Sheet "S4A Fig". Raw luminescence data relevant to S4A Fig. Sheet "S4B Fig". Raw FISH data relevant to S4B Fig. Sheet "Fig 2A". Raw luminescence data relevant to Fig 2A. Sheet "Fig 2B". Raw luminescence data relevant to Fig 2B. Sheet "Fig 2C". Raw luminescence data relevant to Fig 2C. Sheet "Fig 2D". Raw luminescence data relevant to Fig 2D. Sheet "S5A Fig". Raw luminescence data relevant to S5A Fig. Sheet "S5B Fig". Raw luminescence data relevant to S5B Fig. Sheet "Figs 3A, 4A, S6, S9". Raw luminescence data relevant to Figs 3A, 4A, S6, and S9. Sheet "Figs 3B, 4B, S8, S10". Raw luminescence data relevant to Figs 3B, 4B, S8, and S10. Sheet "S7 Fig". Raw luminescence data relevant to S7 Fig. Sheet "Figs 5B, S11". Raw luminescence data relevant to Figs 5B and S11. Sheet "S12 Fig". Raw luminescence data relevant to S12 Fig. Sheet "Fig 6A, 6B". Raw luminescence data relevant to Fig 6A and 6B. Sheet "Fig 7A, 7B". Raw luminescence data relevant to Fig 7A and 7B. Sheet "Fig 8A". Raw luminescence data relevant to Fig 8C. Sheet "Fig 8C". Raw luminescence data relevant to Fig 8C.
(XLSX)

## Acknowledgments

We acknowledge the contributions of Abigail Whalen to the initial development of the riboswitch reporter and advice from other members of our laboratory. We thank Dale Whittington (UW Mass Spectrometry Center) for assistance with the HPLC-MS analysis. We thank Jonathan Jagodnik for helpful discussions about nucleotide-specific riboswitches and Frederico Gueiros Fihlo for comments on the manuscript.

## Author contributions

**Conceptualization:** Molly Hydorn, Sathya Narayanan Nagarajan, Jonathan Dworkin.

**Formal analysis:** Molly Hydorn, Elizabeth Fones.

**Funding acquisition:** Caroline S. Harwood, Jonathan Dworkin.

**Investigation:** Molly Hydorn, Sathya Narayanan Nagarajan, Elizabeth Fones, Caroline S. Harwood.

**Methodology:** Molly Hydorn, Sathya Narayanan Nagarajan, Caroline S. Harwood.

**Project administration:** Caroline S. Harwood, Jonathan Dworkin.

**Supervision:** Caroline S. Harwood.

**Writing – original draft:** Molly Hydorn, Jonathan Dworkin.

**Writing – review & editing:** Molly Hydorn, Sathya Narayanan Nagarajan, Caroline S. Harwood, Jonathan Dworkin.

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
