## [Decision Letter · Decision Letter 0]

29 May 2025

PGENETICS-D-25-00454

Analysis of (p)ppGpp metabolism and signaling using a dynamic luminescent reporter

PLOS Genetics

Dear Dr. Dworkin,

Thank you for submitting your manuscript to PLOS Genetics. After careful consideration, we feel that it has merit but does not fully meet PLOS Genetics's publication criteria as it currently stands. Therefore, we invite you to submit a revised version of the manuscript that addresses the points raised during the review process.

Please submit your revised manuscript within 60 days Jul 28 2025 11:59PM. If you will need more time than this to complete your revisions, please reply to this message or contact the journal office at plosgenetics@plos.org. Please include the following items when submitting your revised manuscript:

We look forward to receiving your revised manuscript.

Kind regards,

Jue D. Wang

Academic Editor

PLOS Genetics

Sean Crosson

Section Editor

PLOS Genetics

Aimée Dudley

Editor-in-Chief

PLOS Genetics

Anne Goriely

Editor-in-Chief

PLOS Genetics

**Journal Requirements:**

At this stage, the following Authors/Authors require contributions: Molly Hydorn, Sathya Narayanan Nagarajan, Elizabeth Fones, Caroline Harwood, and Jonathan Dworkin. Please ensure that the full contributions of each author are acknowledged in the "Add/Edit/Remove Authors" section of our submission form.

The list of CRediT author contributions may be found here: https://journals.plos.org/plosgenetics/s/authorship#loc-author-contributions

4) We notice that your supplementary Figures, and Tables are included in the manuscript file. Please remove them and upload them with the file type 'Supporting Information'. Please ensure that each Supporting Information file has a legend listed in the manuscript after the references list.

5) We note that your Data Availability Statement is currently as follows: "All data is contained within the paper". Please confirm at this time whether or not your submission contains all raw data required to replicate the results of your study. Authors must share the “minimal data set” for their submission. PLOS defines the minimal data set to consist of the data required to replicate all study findings reported in the article, as well as related metadata and methods (https://journals.plos.org/plosone/s/data-availability#loc-minimal-data-set-definition).

- The points extracted from images for analysis..

6) Please amend your detailed Financial Disclosure statement. This is published with the article. It must therefore be completed in full sentences and contain the exact wording you wish to be published. Please ensure that the funders and grant numbers match between the Financial Disclosure field and the Funding Information tab in your submission form. Note that the funders must be provided in the same order in both places as well.

**Reviewers' comments:**

Reviewer's Responses to Questions

**Comments to the Authors:**

Reviewer #1: In this manuscript, Hydron et al. designed and synthesized a riboswitch-based luciferase reporter for (p)ppGpp, the alarmone that senses amino acid limitation, in Bacillus subtilis. Using a multipronged approach, they first confirmed that the luminescence intensity indeed reflects the (p)ppGpp level in cells. Subsequent, they used this reporter to test various aspects of known (p)ppGpp regulation and the results largely agreed with the literature reports except the absence of decrease in GTP pool when (p)ppGpp increases. Overall, I found the reporter very useful – as the authors mentioned in the discussion, riboswitch reporters have now been increasingly used to measure intracellular concentration of second messengers, and there is a need for more reporters for (p)ppGpp in particular. The potential application in mouse model or in natural bacterial communities is also attractive. The manuscript is thorough; the logic is clear; and the assays well designed and executed. I support the publication of the manuscript in PloS Genetics. There are following comments I have on the technical aspect that the authors may wish to consider in a revised manuscript.

Major Comments:

1. Interpretation of (p)ppGpp synthase deletion effects: The authors use a (p)ppGpp⁰ strain (∆relA∆sasA∆sasB) to show that RsFluc luminescence is strongly reduced (Line 114-117, Figure 1B). While this supports the idea that RsFluc primarily reports (p)ppGpp abundance, it may be worth acknowledging that deletion of relA, sasA, and sasB also broadly alters cell physiology, and global changes in transcription or translation could secondarily affect luciferase expression.

2. Timing of luciferase reporter responses: In Figure 2D, RsFluc luminescence (directly sensing (p)ppGpp) and PurE-luc luminescence (reflecting transcriptional activation) are compared. Since both RsFluc and PurE-luc are luciferase-based reporters, I appreciate that their kinetics can be very rapid. However, because PurE-luc reflects transcriptional activation downstream of (p)ppGpp signaling, whereas RsFluc reflects direct sensing of (p)ppGpp abundance, I would still expect a short but noticeable delay in PurE-luc luminescence relative to RsFluc luminescence. In the current figure, the two signals change almost simultaneously (also mentioned in Line 178). Could the authors comment on the expected timing resolution and whether the sampling interval allows detection of a transcriptional delay? Explicitly commenting on this point would help readers better understand the kinetics of (p)ppGpp signaling versus downstream gene expression.

3. OPP labeling interpretation (Figure 8): The OPP labeling experiments nicely demonstrate a decrease in active translation upon (p)ppGpp accumulation. However, I wonder whether additional controls could strengthen the interpretation. In particular:

● Since OPP labeling efficiency and uptake could vary depending on membrane status or cell stress, did the authors assess cell viability/growth or membrane integrity during the OPP labeling period?

● Given that OPP only measures global nascent protein synthesis, is it possible that selective translation of stress-induced proteins persists even when global rates drop?

● Did the authors check whether OPP labeling efficiency is comparable across conditions (wild-type vs. mutant, stressed vs. unstressed)?

● It would improve the clarity by briefly explaining why the authors picked 120 min and 240 min as TP1 and TP2, respectively.

Minor Comments:

1. Figure 1A: The authors should provide more details about the RsFluc construct either in the figure or the figure legend (e.g., Is the red section in the ppGpp riboswitch a RBS?). Also, there is an unintended break or gap in the line within the riboswitch region and a short red line is present immediately after the riboswitch, which does not seem to correspond to any described feature. Suggest correcting these minor artifacts to improve figure clarity.

2. Line 191: Italicizing sasA and sasB in “... or the combination ΔsasAΔsasB ...”.

3. Line 205-207: “…the magnitude of the increase was not substantially affected, indicating that Rel Y200A likely does not impair synthase activity.”. This statement seems only true during a nutrient downshift but not during normal growth. It would increase the clarity by specifying the condition.

4. Line 283: “... as the mupirocin used in the earlier studies rapidly and strongly simulates (p)ppGpp abundance in <5 min.” There is a small typo where "simulates" should be corrected to "stimulates".

5. It would improve the clarity of the bar graphs if individual data points (e.g., scatter overlay) were shown, allowing readers to assess the variability among replicates.

6. While RsFluc activity closely tracks expected (p)ppGpp dynamics, it is worth noting that (p)ppGpp broadly affects transcription, translation, and metabolic remodeling. I suggest briefly discussing the possibility of these indirect effects on reporter expression, acknowledging that while RsFluc predominantly reflects (p)ppGpp levels, secondary physiological changes could modulate reporter expression to a small extent.

Reviewer #2: This study by Hydorn et. al. develops a riboswitch-based luminescent reporter for the alarmone ppGpp that enables temporal monitoring of levels in live B. subtilis cells. Importantly, besides showing that the reporter works through use of mutants (ppGpp-0 strain, synthase knockouts, rel mutants that affect synthase or hydrolase activity), the researchers were able to observe ppGpp changes in response to amino acid downshift and diauxic shift; use riboswitch variants to separately measure pppGpp and ppGpp levels; correlate the riboswitch reporter to promoters known to be activated by ppGpp. The riboswich-based reporter can be an important tool for other researchers in the field who have sought to study ppGpp dynamics in B. subtilis; however I have the following questions about the study:

1. The researchers do not provide the sequence of their promoter-riboswitch-luciferase reporter construct, they only show a diagram of the construct in Fig. 1A. This will make it difficult for other researchers to make use of this tool.

2. The sequence of the double riboswitch mutant is not provided, and data on this mutant appear to show that it reduces the signal but does not fully disrupt the riboswitch response. Is there a more "null" riboswitch mutant that could be used?

3. Given the strong effect of ppGpp on translation, which the researchers also demonstrate in Figs 8 and S11, they should address whether the drop observed in RLU/OD signal after the peak that typically is at ~240 min may be due to ppGpp decreasing reporter protein synthesis, rather than an actual drop in ppGpp levels. How is the reporter not going to be affected by reduced protein synthesis if its half-life is ~5 mins?

4. Some additional comments: (i) the authors should discuss the ATP and oxygen dependence of the firefly luciferase reporter, which may complicate imaging (ii) "the first time" claim in summary - please see 10.1002/anie.202111170; (iii) "adds them to the 3' carbon" - should be 3' hydroxyl

Reviewer #3: ppGpp is a profoundly important signaling molecule used by diverse bacterial species. In this study, the authors built a biosensor using a ppGpp-sensing riboswitch, thereby allowing them to investigate the relative intracellular dynamics of ppGpp. They validated this biosensor through a collection of mutant strains and growth conditions. Additionally, the authors demonstrated that the ppGpp-sensing reporter fusion could be used to gain insight into the functional role(s) and significance of different synthase and hydrolase enzymes. The manuscript is clear and the data overall supports the authors’ claims. I am confident this manuscript will be well-cited and that other researchers will quickly adopt usage of the ppGpp riboswitch reporter fusion. I have only a few minor comments.

Line 72: remove PMID number.

Lines 473-4: An additional sentence including brief description of culture medium (or a reference) would improve this methods subsection.

Fig 2B: Is it necessary to include the key if the labels will also be included on the x-axis? Also, since “RsFluc-mut” and “M9+M11” refer to the same reporter, choosing one nomenclature will make it easier to understand for the reader. Similarly, a different nomenclature style is used in panel 2D (“ppGpp RS-luciferase”), so coordinating the labels across the figures would help the reader.

Fig 2C: The spike in ppGpp signal as the bacterium switches carbon sources during diauxic growth in interesting; however, since the flattening of the growth curve is essentially a temporary stationary phase, wouldn’t this result imply that a spike in ppGpp would coincide with depletion of the glucose even when a second carbon source had not been included?

Fig 3A: it is not necessary to include any additional experimentation in this figure; however, while the main text points out that the sasA/B double mutant is defective for ppGpp after amino acid downshift, the double mutant is not shown in Fig 3A for comparison, instead it is shown in Fig S5. The data can all be found in the current manuscript, and it therefore does not require any experimental changes. That being said, the location of the data in S5 is a bit odd.

Fig 8A: error measurements are included for the luciferase reporter but not for the growth data. It is a minor comment, but is there a specific reason for that?

**Have all data underlying the figures and results presented in the manuscript been provided?**

Reviewer #1: Yes

Reviewer #2: **No: **Sequence information and numerical data spreadsheets not found

Reviewer #3: None

PLOS authors have the option to publish the peer review history of their article (what does this mean?). If published, this will include your full peer review and any attached files.

Reviewer #1: No

Reviewer #2: No

Reviewer #3: No

**Figure resubmission:**
---

## [Decision Letter · Decision Letter 1]

29 Jul 2025

PGENETICS-D-25-00454R1

Analysis of (p)ppGpp metabolism and signaling using a dynamic luminescent reporter

PLOS Genetics

Dear Dr. Dworkin,

Thank you for submitting your manuscript to PLOS Genetics. After careful consideration, we feel that it has merit but does not fully meet PLOS Genetics's publication criteria as it currently stands. Therefore, we invite you to submit a revised version of the manuscript that addresses the points raised during the review process.

Please submit your revised manuscript within 30 days Aug 28 2025 11:59PM. If you will need more time than this to complete your revisions, please reply to this message or contact the journal office at plosgenetics@plos.org. Please include the following items when submitting your revised manuscript:

We look forward to receiving your revised manuscript.

Kind regards,

Jue D. Wang

Academic Editor

PLOS Genetics

Sean Crosson

Section Editor

PLOS Genetics

Aimée Dudley

Editor-in-Chief

PLOS Genetics

Anne Goriely

Editor-in-Chief

PLOS Genetics

**Additional Editor Comments :**

Hi Jonathan,

As you can see the reviewers are in general happy about your revision. However, if you can address the additional minor comments from reviewer 2, that would be great.

**Journal Requirements:**

1) Please ensure that the CRediT author contributions listed for every co-author are completed accurately and in full.
At this stage, the following Authors/Authors require contributions: Molly Hydorn, Sathya Narayanan Nagarajan, Elizabeth Fones, Caroline Harwood, and Jonathan Dworkin. Please ensure that the full contributions of each author are acknowledged in the "Add/Edit/Remove Authors" section of our submission form.
The list of CRediT author contributions may be found here: https://journals.plos.org/plosgenetics/s/authorship#loc-author-contributions
 
2) We note that your Supplementary Tables files are duplicated on your submission. Please remove them from the main file of the manuscript as they should only be uploaded separately with the file type 'Supporting Information'
 
3) We do not publish any copyright or trademark symbols that usually accompany proprietary names, eg ©,  ®, or TM  (e.g. next to drug or reagent names). Therefore please remove all instances of trademark/copyright symbols throughout the text, including:
- ® : Fluor® on page 21.
 
4) We have noticed that you have uploaded Supporting Information files, but you have not included a complete list of legends. Please add a full list of legends for your Supporting Information file (compiled raw data copy.xlsx .) after the references list.
 
5) Please amend your detailed Financial Disclosure statement. This is published with the article. It must therefore be completed in full sentences and contain the exact wording you wish to be published.
1) State the initials, alongside each funding source, of each author to receive each grant. For example: "This work was supported by the National Institutes of Health (####### to AM; ###### to CJ) and the National Science Foundation (###### to AM)."
2) State what role the funders took in the study. If the funders had no role in your study, please state: "The funders had no role in study design, data collection and analysis, decision to publish, or preparation of the manuscript."
3) If any authors received a salary from any of your funders, please state which authors and which funders..
 
6) Please provide a completed 'Competing Interests' statement, including any COIs declared by your co-authors. If you have no competing interests to declare, please state "The authors have declared that no competing interests exist". Otherwise please declare all competing interests beginning with the statement "I have read the journal's policy and the authors of this manuscript have the following competing interests:"

**Reviewers' comments:**

Reviewer's Responses to Questions

Reviewer #1: The authors have addressed all my concerns. I recommend the publication of the manuscript.

Reviewer #2: The authors have addressed most of my prior review concerns and questions, however there are some minor aspects that remain:

R2 Point #3: "Second, while (p)ppGpp has a clear inhibitory effect on global protein synthesis (54), this effect would tend to dampen the response of a protein reporter such as luciferase thereby minimizing observed (p)ppGpp-dependent increase in RsFluc activity." The original review point was not about the observed luc signal spike, which this reviewer did not doubt is due to a rise in ppGpp, but instead whether the subsequent DROP in luciferase signal could be due to ppGpp-dependent inhibition of global protein synthesis. Meaning, ppGpp levels might remain high for longer than is observed by the luc reporter. Notably, the ppGpp-0 control cannot rule this out. In the revision, to address this experimentally, the authors have included quantitation of the reporter mRNA by FISH, shown in Fig. S4. However, they need to address the following issues:

(a) Fig. S4A - the x-axis is missing the timepoint values. In particular, without knowing the time difference between TP3 and TP4, it is not possible to evaluate whether the mRNA FISH data make sense with the reporter luminescence data. These results also bear more discussion, because they suggest that ppGpp levels are already low at TP4, which is when luc reporter signal is highest, so there appears to be a time-lag that should be noted. Also, ppGpp levels dropping should not affect the stability of existing full-length transcripts, so depending on the time difference between TP3 and TP4, the observed FISH result may be surprising.

(b) Description of new FISH experiment in the Methods section does not describe how the fluorescence was quantitated. The level of statistical also is not described (how many cells; how many probes).

**Have all data underlying the figures and results presented in the manuscript been provided?**

Reviewer #1: Yes

Reviewer #2: Yes

PLOS authors have the option to publish the peer review history of their article (what does this mean?). If published, this will include your full peer review and any attached files.

Reviewer #1: No

Reviewer #2: No

**Figure resubmission:**
---

## [Decision Letter · Decision Letter 2]

11 Aug 2025

Dear Dr Dworkin,

We are pleased to inform you that your manuscript entitled "Analysis of (p)ppGpp metabolism and signaling using a dynamic luminescent reporter" has been editorially accepted for publication in PLOS Genetics. Congratulations!

Yours sincerely,

Jue D. Wang

Academic Editor

PLOS Genetics

Sean Crosson

Section Editor

PLOS Genetics

Aimée Dudley

Editor-in-Chief

PLOS Genetics

Anne Goriely

Editor-in-Chief

PLOS Genetics

Comments from the reviewers (if applicable):

Reviewer's Responses to Questions

**Comments to the Authors:**

Reviewer #2: The authors have addressed my remaining feedback in their second revision.

**Have all data underlying the figures and results presented in the manuscript been provided?**

Reviewer #2: Yes

PLOS authors have the option to publish the peer review history of their article (what does this mean?). If published, this will include your full peer review and any attached files.

Reviewer #2: No

**Data Deposition**

http://datadryad.org/submit?journalID=pgenetics&manu=PGENETICS-D-25-00454R2

**Press Queries**

---

## [Editor Report · Acceptance letter]

PGENETICS-D-25-00454R2

Analysis of (p)ppGpp metabolism and signaling using a dynamic luminescent reporter

Dear Dr Dworkin,

We are pleased to inform you that your manuscript entitled "Analysis of (p)ppGpp metabolism and signaling using a dynamic luminescent reporter" has been formally accepted for publication in PLOS Genetics! Your manuscript is now with our production department and you will be notified of the publication date in due course.

With kind regards,

Anita Estes

PLOS Genetics

On behalf of:
